# Learning Accurate Entropy Model with Global Reference for Image Compression

**Yichen Qian**
Alibaba Group
Hangzhou, China
yichen.qyc@alibaba-inc.com

**Zhiyu Tan**
Alibaba Group
Hangzhou, China
zhiyu.tzy@alibaba-inc.com

**Xiuyu Sun**[*]
Alibaba Group
Hangzhou, China
xiuyu.sxy@alibaba-inc.com

**Ming Lin**
Alibaba Group
Bellevue, WA, 98004, USA
ming.l@alibaba-inc.com

**Dongyang Li**
Alibaba Group
Hangzhou, China
yingtian.ldy@alibaba-inc.com

**Zhenhong Sun**
Alibaba Group
Hangzhou, China
zhenhong.szh@alibaba-inc.com

**Hao Li**
Alibaba Group
Hangzhou, China
lihao.lh@alibaba-inc.com

**Rong Jin**
Alibaba Group
Bellevue, WA, 98004, USA
jinrong.jr@alibaba-inc.com

## Abstract

In recent deep image compression neural networks, the entropy model plays a critical role in estimating the prior distribution of deep image encodings. Existing methods combine hyperprior with local context in the entropy estimation function. This greatly limits their performance due to the absence of a global vision. In this work, we propose a novel Global Reference Model for image compression to effectively leverage both the local and the global context information, leading to an enhanced compression rate. The proposed method scans decoded latents and then finds the most relevant latent to assist the distribution estimating of the current latent. A by-product of this work is the innovation of a mean-shifting GDN module that further improves the performance. Experimental results demonstrate that the proposed model outperforms the rate-distortion performance of most of the state-of-the-art methods in the industry.

## 1 Introduction

Image compression is a fundamental research topic in computer vision. The goal of image compression is to preserve the critical visual information of the image while reducing the bit-rate for storage or transmission. The state-of-the-art image compression standards, such as JPEG (Wallace, 1992), JPEG2000 (Rabbani & Joshi, 2002), HEVC/H.265 (Sullivan et al., 2012) and Versatile Video Coding (VVC) (Ohm & Sullivan, 2018), are carefully engineered and highly tuned to achieve better performance.

Albeit widely deployed, the conventional human-designed codecs take decades of development to achieve impressive compression rate today. Any further improvement is expected to be even more difficult. Inspired by the successful stories of deep learning in many vision tasks, several pioneer works (Toderici et al., 2016; Agustsson et al., 2017; Theis et al., 2017; Ballé et al., 2017; Ballé et al., 2018; Mentzer et al., 2018; Lee et al., 2019; Minnen et al., 2018a) demonstrate that the image compression task can be effectively solved by deep learning too. This breakthrough allows us to use data-driven learning system to design novel compression algorithms automatically. As a result, a majority of deep image compression (DIC) models are based on autoencoder framework. In this framework, an encoder transforms pixels into a quantized latent representation suitable for compression, while a decoder is jointly optimized to transform the latent representation back into pixels.

---

[*]Corresponding author.

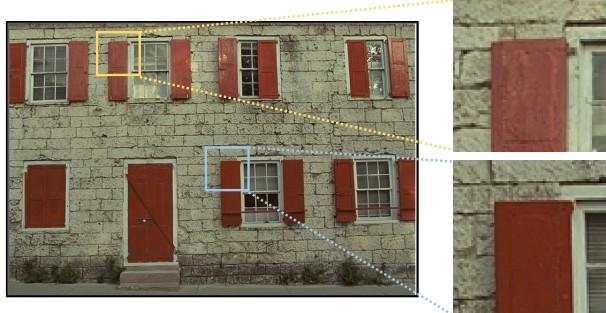

Figure 1: Global spatial redundancy in the image. For standard codecs and previous learned codecs, non-local relevant patches (marked by yellow and blue) would consume equal bit rates.

The latent representation can be losslessly compressed to create a bitstream by using entropy coding method (Rissanen & Langdon, 1981).

In the entropy coding, the compression quality is controlled by the entropy estimation of latent features generated by the encoder. It is therefore important to learn an accurate entropy model. To this end, several solutions have been considered. With additional bits, some methods propose entropy model conditioned on a hyperprior, using side information of local histograms over the latent representation (Minnen et al., 2018b) or a hierarchical learned prior (Ballé et al., 2018). Context-adaptive models (Minnen et al., 2018a; Lee et al., 2019) incorporate predictions from neighboring symbols to avoid storing the additional bits. While these methods improve the accuracy of the entropy models, they are unable to use global context information during the compression, leading to suboptimal performance.

In this work, we observe that global spatial redundancy remains in the latents, as shown in Figure 1. Motivated by this, we propose to build up a global relevance throughout the latents. Inspired by the recent reference-based Super-Resolution (SR) methods (Zheng et al., 2018; Yang et al., 2020), we empower the entropy model with global vision by incorporating a reference component. Unlike the super-resolution scenario, incorporating global reference information is non-trivial in deep image compression. The image during decoding is often incomplete which means that the information is badly missing. Besides, our target is to reduce the bit rates and recover the image from the bitstream faithfully, rather than inpainting a low-resolution image with vivid generated details.

To address the above challenges, in our proposed method, a global reference module searches over the decoded latents to find the relevant latents to the target latent. The feature map of the relevant latent is then combined with local context and hyperprior to generate a more accurate entropy estimation. A key ingredient in the global reference ensemble step is that we consider not only the similarity between the relevant and the target but also a confidence score to measure the high-order statistics in the latent feature distribution. The introduction of the confidence score enhances the robustness of the entropy model, especially for images with noisy backgrounds. Also, we found that the widely used Generalized Divisive Normalization (GDN) in image compression suffers from a mean-shifting problem. Since the GDN densities are zero-mean by definition, mean removal is necessary to fit the density (Ballé et al., 2016b). Therefore we propose an improved version of GDN, named GSDN (Generalized Subtractive and Divisive Normalization) to overcome this difficulty.

We summarize our main contributions as follows:

- To the best of our knowledge, we are the first to introduce global reference into the entropy model for deep image compression. We develop a robust reference algorithm to ensemble local context, global reference and hyperprior in a novel architecture. When estimating the latent feature entropy, both similarity score and confidence score of the reference area are considered to battle with the noisy background signals.
- We propose a novel GSDN module that corrects the mean-shifting problem.
- Experiments show that our method outperforms the most advanced codes available today on both PSNR and MS-SSIM quality metrics. Our method saves by 6.1% compared to the context-adaptive deep models (Minnen et al., 2018b; Lee et al., 2019) and as much as 21.0% relative to BPG (Bellard., 2014).

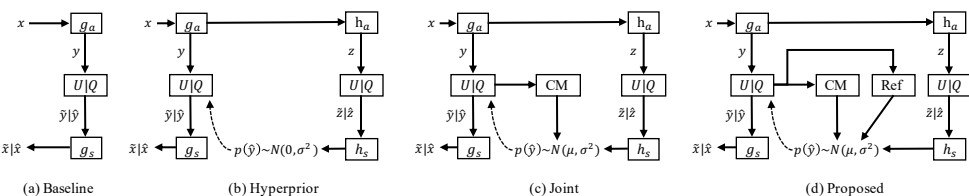

Figure 2: Operational diagrams of learned compression models (a)(b)(c) and proposed Reference-based Entropy Model (d).

The remainder of this work is organized as follows. In Section 2, we introduce the backbone of the end-to-end deep image compression network as well as the reference-based component for the entropy model. Section 3 demonstrates the structure of our combined entropy model. The GSDN with mean-shifting correction is given in Section 4. We present experimental comparison and visualization in Section 5. Finally, we enclose this work with an open discussion in Section 6.

## 2 LEARNED IMAGE COMPRESSION

Learned image compression using deep neural networks has attracted considerable attention recently. The work of Toderici et al. (2016) first explored a recurrent architecture using an LSTM-based entropy model. A wide range of models (Ballé et al., 2017; Ballé et al., 2018; Mentzer et al., 2018; Minnen et al., 2018a; Lee et al., 2019; Hu et al., 2020; Cheng et al., 2020) used a CNN-based autoencoder with constrained entropy.

General learned image compression consists of an encoder, a quantizer, a decoder, and an entropy model. An image $x$ is transformed into a latent representation $y$ via the encoder $g_a(x)$, which is discretized by the quantizer $Q(y)$ to form $\hat{y}$. Given the entropy model $p_{\hat{y}}$, the discretized value $\hat{y}$ can be compressed into a bitstream using entropy coding techniques such as arithmetic coding (Rissanen & Langdon, 1981). The decoder $g_s(\hat{y})$ then forms the reconstructed image $\hat{x}$ from the quantized latent representation $\hat{y}$, which is decompressed from the bitstream. The training goal for learned image compression is to optimize the trade-off between the estimated coding length of the bitstream and the quality of the reconstruction, which is a rate-distortion optimization problem:

$$\mathcal{L} = R + \lambda D = \mathbb{E}_{x \sim p_x}[-\log_2 p_{\hat{y}}(Q(g_a(x)))] + \lambda \mathbb{E}_{x \sim p_x}[d(x, g_s(\hat{y})], \tag{1}$$

where $\lambda$ is the coefficient which controls the rate-distortion trade-off, $p_x$ is the unknown distribution of natural images. The first term represents the estimated compression rate of the latent representation. The second term $d(x, \hat{x})$ represents the distortion value under given metric, such as mean squared error (MSE) or MS-SSIM (Wang et al. (2003)).

Entropy coding relies on an entropy model to estimate the prior probability of the latent representation. Ballé et al. (2017) propose a fully factorized prior for entropy estimation as shown in Figure 2(a), while the prior probability of discrete latent representations is not adaptive for different images. As shown in Figure 2(b), Ballé et al. (2018) model the latent representation as a zero-mean Gaussian distribution based on a spatial dependency with additional bits. In Lee et al. (2019) and Minnen et al. (2018a), they introduce an autoregressive component into the entropy model. Taking advantage of high correlation of local dependency, context-adaptive models contribute to more accurate entropy estimation. However, since their context-adaptive entropy models only capture the spatial information of neighboring latents, there is redundant spatial information across the whole image. To further remove such redundancy, our method incorporates a reference-based model to capture global spatial dependency.

Specially for learned image compression, a generalized divisive normalization (GDN) (Ballé et al., 2016a) transform with optimized parameters has proven effective in Gaussianizing the local joint statistics of natural images. Unlike many other normalization methods whose parameters are typically fixed after training, GDN is spatially adaptive therefore is highly nonlinear. As the reference-based model calculates relevance over the latents, it is crucial to align the distribution of the latents. To better align the latents, the proposed GSDN incorporates a subtracting factor with GDN. We also present an effective method of inverting it when decompressing the latent representation back to image.

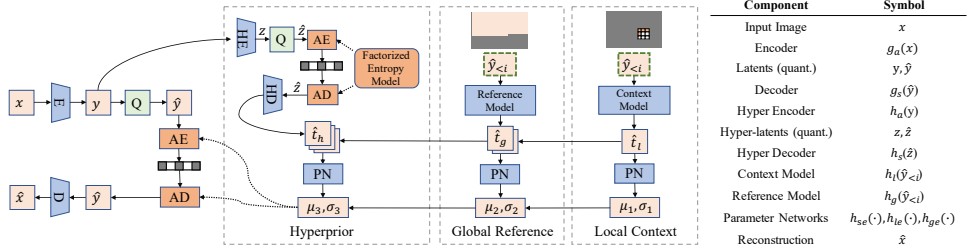

Figure 3: Our compression model with combined local, global, and hyperprior entropy model. Tan items represent data tensors, blue represents learned modules (*e.g.* convolutional layer), green is for quantization, and red represents entropy coding. The left side shows an autoencoder with a quantizer, the right side corresponds to the entropy model. The entropy model progressively incorporates local context, global context and hyperprior. Each parameter network predicts the Gaussian parameters conditioned on both the previous features and predictions. Using the Gaussian parameters $(\mu_3, \sigma_3)$, the quantized latents are compressed into a stream by an arithmetic encoder (AE) and decompressed by an arithmetic decoder (AD).

## 3    COMBINED LOCAL, GLOBAL AND HYPERPRIOR ENTROPY MODEL

The models we analyze in this paper build on the architecture introduced in Minnen et al. (2018a), which combined an autoregressive model with the hyperprior. Figure 3 provides a high-level overview of our approach. The compression model contains two main sub-networks. The first is the core autoencoder, which learns the transform and the inverse transform between image and latent representation. $Q$ represents the quantization function. The gradient-based optimization in learned methods is hindered by quantization. Here, we make use of a mixed approach that has proven efficient in Minnen & Singh (2020). The second sub-network is the combined entropy model, which is responsible for estimating a probabilistic model over the latents for entropy coding. The combined entropy model consists of a context model, a reference model, and a hyper-network (hyper encoder and hyper decoder). The three components are combined progressively. Then three parameter networks generate the mean and scale parameters for a conditional Gaussian entropy model respectively.

Following the work of Minnen et al. (2018a), we model each latent, $\hat{y}_i$, as a Gaussian with mean $\mu_i$ and deviation $\sigma_i$ convolved with a unit uniform distribution:

$$p_{\hat{y}}(\hat{y}|\hat{z}, \theta) = \prod_{i=1} (\mathcal{N}(\mu_i, \sigma_i^2) * \mathcal{U}(-0.5, 0.5))) (\hat{y}_i) \tag{2}$$

where $\mu$ and $\sigma$ are the predicted parameters of entropy model, $\hat{z}$ is the quantized hyper-latents, $\theta$ is the entropy model parameters. The entropy model for the hyperprior is the same as in Ballé et al. (2018), which is a non-parametric, fully factorized density model. As the hyperprior is part of the compressed bitstream, we extend the rate of Equation 1 as follows:

$$R = \mathbb{E}_{x \sim p_x}[-\log_2 p_{\hat{y}}(\hat{y})] + \mathbb{E}_{x \sim p_x}[-\log_2 p_{\hat{z}}(\hat{z})] \tag{3}$$

The compressed latents and the compressed hyper-latents are part of the bitstream.

The reference-based SR methods (Zheng et al., 2018; Yang et al., 2020) adopt "patch match" to search for proper reference information. However, in the serial processing of image compression, the latent representation during decoding is often incomplete. We extend this search method by using a masked patch. Figure 4 illustrates how the relevance embedding module estimates similarity and fetches the relevant latents. When decoding the target latent, we use neighboring latents (left and top) as a basis to compute the similarities between the target latent and its previous latents. Particularly, the latents are unfolded into patches and then masked, denoted as $q \in [H \times W, k \times k \times C]$ (where $H$, $W$, $k$, $C$ correspond to height, width, unfold kernel size and channels, respectively). We calculate the similarity matrix $r \in [H \times W, H \times W]$ throughout the masked patches by using cosine similarity,

$$r_{i,j} = \left\langle \frac{q_i}{\|q_i\|}, \frac{q_j}{\|q_j\|} \right\rangle \tag{4}$$

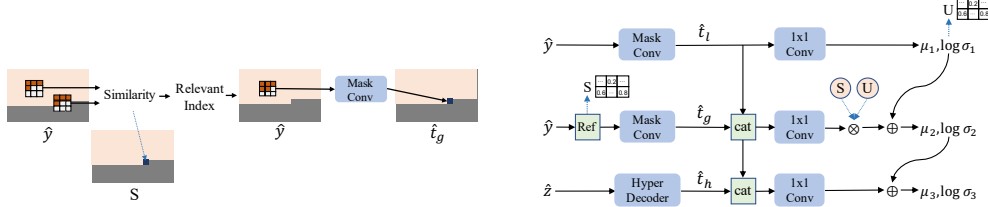

Figure 4: A mask slide patch searches on all the decoded latents (tan area). The relevant latents are fetched and learned with a masked convolution.

Figure 5: The progressive entropy model incorporates three sub-models. The reference model is a soft-attention-like function.

Note that we can only see the decoded latents, so the lower triangular of the similarity matrix is set to zero. We get the most relevant position for each latent as well as the similarity score. According to the position, we fetch the neighboring latents (left and top) as well as the center latent, which is named as "relevant latents". We use a masked convolution as in Van den Oord et al. (2016) to transfer the relevant latents.

To measure how likely a reference patch perfectly matches the target patch, Yang et al. (2020) propose a soft-attention module to transfer features by using the similarity map $S$. However, we found that a similarity score is not sufficient to reflect the quality of reference latent in image compression. For this reason, a confidence score is introduced to measure the texture complexity of the relevant latent. We use the context model to predict the Gaussian parameters (*i.e.*, $\mu_1, \sigma_1$) of latents solely. The latents $\hat{y}$ are now modeled as Gaussian with mean $\mu_1$ and standard deviation $\sigma_1$. The probabilities of the latents are then calculated according to $(\mu_1, \sigma_1)$ as in Equation 2. As reference model is designed in spatial dimension, the confidence map $U$ is obtained by averaging the probabilities across channel. With the above two parameters, the more relevant latent combination would be enhanced while the less relevant one would be relived. The similarity $S$ and the confidence $U$ are both 2D feature maps.

Figure 5 provides the structure of our combined entropy model. For the context model, we transfer the latents (*i.e.*, $\hat{y}$) with a masked convolution. For the reference model, we transfer the unfolded relevant latents with a masked convolution. We use $1 \times 1$ convolution in the parameter networks. Local, global and hyperprior features are ensembled stage by stage, as well as the predicted Gaussian parameters. The mean parameters are estimated by the context model first, and then updated by the global model and the hyperprior model. We use the Log-Sum-Exp trick for resolving the under or overflow issue of deviation parameters. The output of global reference is further multiplied by the similarity $S$ and the confidence $U$. The context model is based on the neighboring latents of the target latent to reduce the local redundancy. From the perspective of the global context, the reference model makes further efforts to capture spatial dependency. As the first two models predict by the decoded latents, there exists uncertainty that can not be eliminated solely. The hyperprior model learns to store information needed to reduce the uncertainty. This progressive mechanism allows an incremental accuracy for distribution estimation.

## 4 GENERALIZED SUBTRACTIVE AND DIVISIVE NORMALIZATION

Virtually the traditional image and video compression codec consists of several basic modules, *i.e.* transform, quantization, entropy coding and inverse transform. An effective transform for image compression maps from the image to a compact and decorrelated latent representation. As part of a Gaussianizing transformation, a generalized divisive normalization (GDN) joint nonlinearity has proven effective at removing statistical dependencies in image data (Ballé et al., 2016b). It shows an impressive capacity for learned image compression.

We define a generalized subtractive and divisive normalization (GSDN) transform that incorporates a subtractive operation. Inspired by the zero-mean definition of Gaussian density, an adaptive subtractive operation is applied before the divisive operation. Particularly, we apply subtractive-divisive normalization after convolution and subsampling operation in the encoder $g_a$ (except the last convolution layer). We represent the $i$th channel at a spatial location as $u_i$. The normalization operation

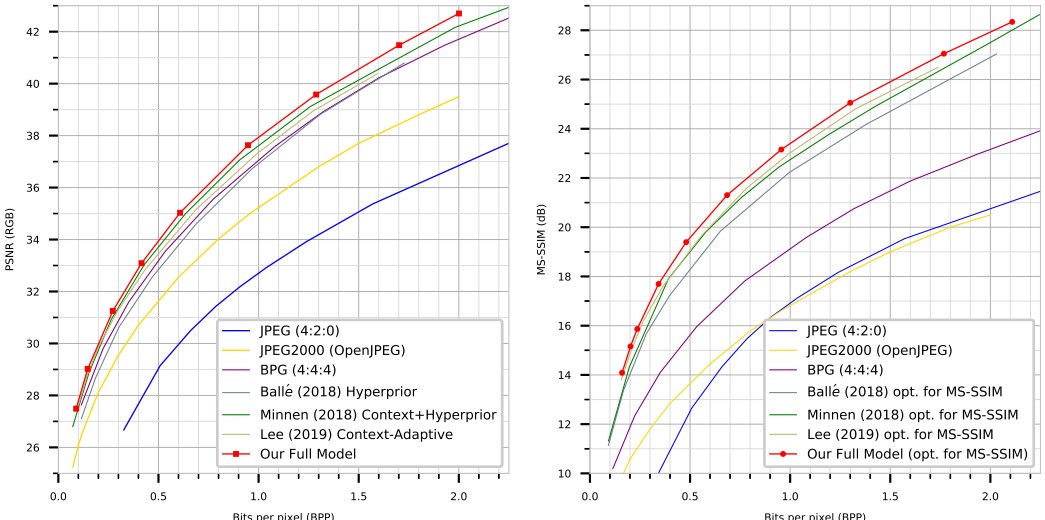

Figure 6: Rate–distortion curves aggregated over the Kodak dataset. The left plot shows peak signal-to-noise ratios as a function of bit rate ($10 \log_{10} \frac{255^2}{d}$ , with $d$ representing mean squared error), the right plot shows MS-SSIM values converted to decibels ($-10 \log_{10}(1 - d)$, where $d$ is the MS-SSIM value in the range between zero and one). In both terms, our full model consistently outperforms standard codecs and the state-of-the-art learned models.

is defined by:

$$w_i = \frac{u_i - (\boldsymbol{\nu}_i + \sum_j \boldsymbol{\tau}_{ij} u_j)}{(\boldsymbol{\beta}_i + \sum_j \boldsymbol{\gamma}_{ij} u_j^2)^{\frac{1}{2}}} \tag{5}$$

The parameter consists of two vectors ($\boldsymbol{\beta}$ and $\boldsymbol{\nu}$) and two matrices ($\boldsymbol{\gamma}$ and $\boldsymbol{\tau}$), for a total of $2 \times (N + N^2)$ parameters (where N is the channels of input feature). The normalization operation shares parameters across the spatial dimension.

We invert the normalization operation in the decoder $g_s$ based on the inversion of GDN introduced in Ballé et al. (2016a). We apply inverse GSDN (IGSDN) after deconvolution and upsampling operation (except the last deconvolution layer) correspond to the encoder. In the decoder, we represent the $i$th channel at a spatial location as $\hat{w}_i$. For the inverse solution, subtraction is replaced by addition while division is replaced by multiplication:

$$\hat{u}_i = \hat{w}_i \cdot (\hat{\boldsymbol{\beta}}_i + \sum_j \hat{\boldsymbol{\gamma}}_{ij} \hat{w}_j^2)^{\frac{1}{2}} + (\hat{\boldsymbol{\nu}}_i + \sum_j \hat{\boldsymbol{\tau}}_{ij} \hat{w}_j) \tag{6}$$

## 5 EXPERIMENTS

### 5.1 IMPLEMENTATION DETAILS

**Architecture** For the results in this paper, we did not make efforts to reduce the capacity (*i.e.* number of channels, layers) of the artificial neural networks to optimize computational complexity. The architecture of our approach extends on the work of Minnen et al. (2018a) in two ways. First, the main autoencoder is extended by replacing the GDN with the proposed GSDN (IGDN with IGSDN in decoder). Second, the entropy model is extended by incorporating a reference model. Three modules are combined in progressively.

**Training** The models were trained on color PNG images from CVPR workshop CLIC training dataset (http://challenge.compression.cc/). The models were optimized using Adam (Kingma & Ba, 2014) with a batch size of 8 and a patch size of $512 \times 512$ randomly extracted from the training dataset. Note that large patch size is necessary for the training of the reference model. As our combined entropy model have three predictive Gaussian parameters, we first trained the three modules with weight of $0.3 : 0.3 : 0.4$ as a warm-up with 1000 epochs. After that, we

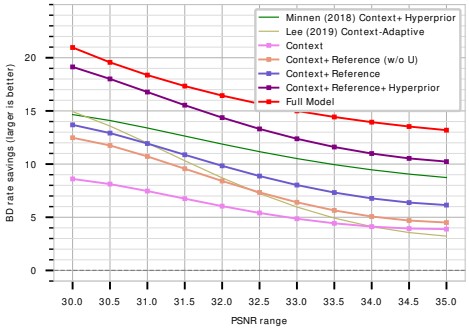

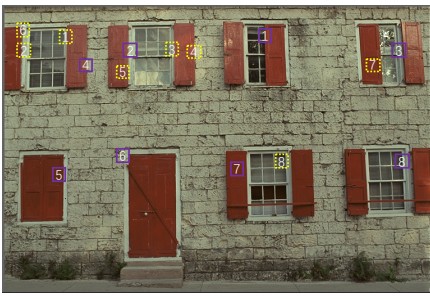

Figure 7: Each curve shows the rate savings at different PSNR quality levels relative to BPG. Our full model outperforms BPG by 21% at low bit rates.

Figure 8: Examples of target region (indicated by purple) and its relevant region (indicated by yellow).

trained three modules with weight of $0.1 : 0.1 : 0.8$ because the third output is used for entropy coding in practice. In the experiments, we trained different models with different $\lambda$ to evaluate the rate-distortion performance for various ranges of bit-rate.

**Distortion measure**     We optimized the networks using two different types of distortion terms, one with MSE and the other with MS-SSIM (Wang et al., 2003). For each distortion type, the average bits per pixel (BPP) and the distortion, PSNR and MS-SSIM, over the test set are measured for each model configurations.

**Other codecs**     For the standard codecs, we used BPG (Bellard., 2014) and JPEG (Wallace, 1992). For the learning-based codecs, we compared state-of-the-art methods that combine spatial context with a hyperprior (Minnen et al. (2018a); Lee et al. (2019)), which also shares a similar structure of our method.

## 5.2   RATE DISTORTION PERFORMANCE

We evaluate the effects of global reference and GSDN in learned image compression. Figure 6 shows RD curves over the publicly available Kodak dataset (Kodak, 1993) by using peak signal-to-noise ratio (PSNR) and MS-SSIM as the image quality metric. The RD graphs compare our full model (Entropy Model with Reference + GSDN) to existing image codecs. In terms of PSNR and MS-SSIM, our model shows better performance over state-of-the-art learning-based methods as well as standard codecs.

Figure 7 shows the results that compares different versions of our models. Particularly, it plots the rate savings for each model relative to the curve for BPG. This visualization provides a more readable way than a standard RD graph. It shows that the four components (*i.e.* local context, global reference, hyperprior) yield progressive improvement. Especially, the combination of global reference provides a rate saving of 5.3% over the context-only model at the low bit rates. As we introduce the confidence $U$, the performance of the reference model is further improved. Our full model, which replaces GDN with GSDN, provides a rate savings about 2.0% over the proposed entropy model.

## 5.3   VISUAL RESULTS OF REFERENCE-BASED ENTROPY MODEL AND GSDN

Figure 8 shows the results of the relevance embedding module. The target region (indicated by purple) and its relevant region (indicated by yellow) are marked by the same numbers. As the relevance is calculated on the latents, we map the position back to the RGB image. The region box just indicates position of target latent (relevant latent) and does not represent receptive field. The relevance results explain the bit savings caused by the combining of global reference.

Figure 9 visualizes the internal mechanisms of different entropy model variants. Three variants are showed: local context only (first row), combined global reference with local context (second row), full entropy model(third row). Intuitively, we can see how these components are complemen-

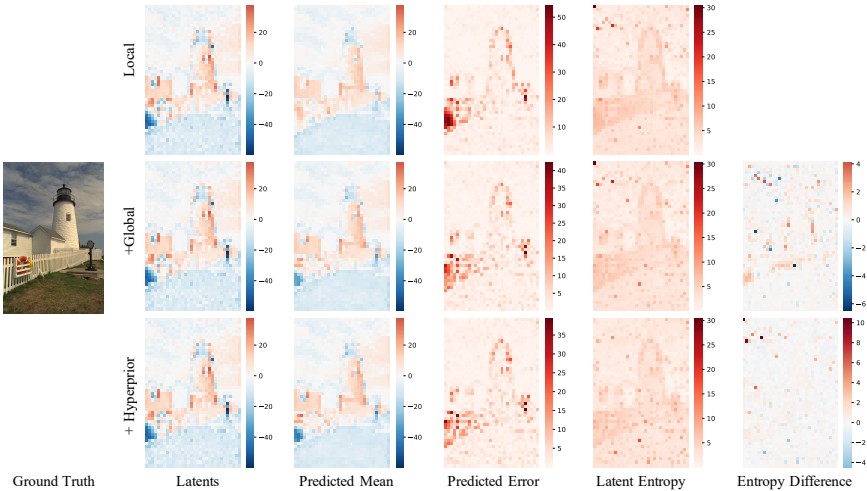

Figure 9: Each row corresponds to a different entropy model variant and shows information for the channel with the highest entropy. The predicted mean corresponds to the Gaussian parameters (*i.e.* $\mu_1$, $\mu_2$, $\mu_3$). The last column shows progressive entropy difference (red is better while blue is worse). The visualizations show that the combined model reduces the prediction error. The ensemble of local context, global reference and hyperprior allows a more accurate estimation and thus a lower entropy.

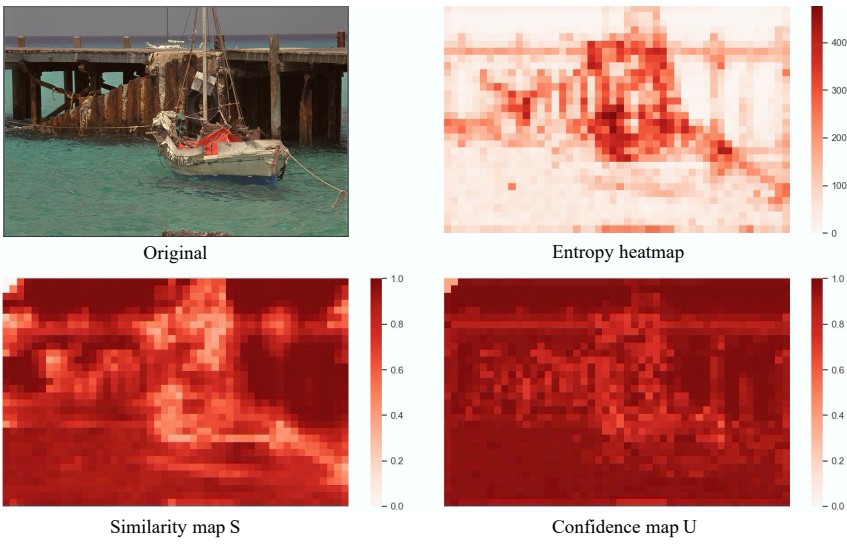

Figure 10: Example of confidence U map and similarity S map. The S map is tend to represent shape while the U map is tend to represent texture, e.g., the wood pile on the right of the boat.

tary. The Kodak image 19 (first column) is encoded, and then the latents for the channel with the highest entropy (second column) are extracted. The predicted mean (third column) corresponds to the Gaussian parameters (*i.e.* $\mu_1$, $\mu_2$, $\mu_3$) estimated by the three entropy model variants. Since the context model is based on the prediction from casual context, it has limited accuracy when dealing with irregular texture.The reference model can search throughout latents that have been decoded and benefit from similar texture. The combining of global reference over local context leads to a lower prediction error. Finally, the last two columns show how the entropy is distributed across the image for the latents. The global reference leads to rate savings over the latents which have similar reference latents, especially those with irregular texture. From the perspective of hyperprior, it uses bit-consuming side information by which the uncertainty could be reduced over all the latents.

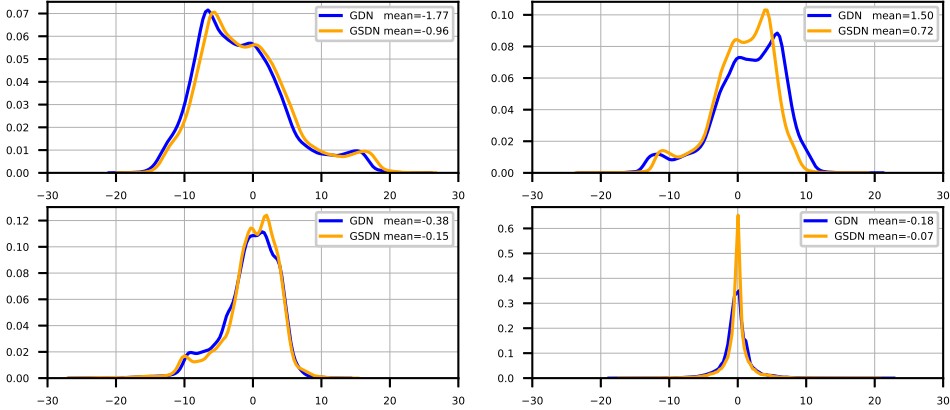

Figure 11: Histograms of the latent representation by GDN-based model and GSDN-based model. Each plot corresponds to one channel of the latents over 24 Kodak images. Four channels with highest entropy are visualized.

As shown in Figure 10, the $S$ map is tend to represent shape while the $U$ map is tend to represent texture, *e.g.*, the wood pile on the right of the boat. The multiplication by the confidence $U$ would influence the similarity $S$. The $U$ map represents the texture score of the reference latent, which measures the complexity of the reference feature. It agrees with what we have assumed that $U$ would compensate for the weakness of $S$.

We also visualized the distributions of the latents on the GDN-based model as well as the GSDN-based model. We constructed a two-stage model. A GDN-based model is trained first, which is then used to construct a GSDN-based model by adding parameters of subtraction. Histograms of the latent representation are shown in Figure 11. Compared to GDN, GSDN relieves the mean-shifting problem of the latents. The latents of GSDN-based model comes closer to zero-mean.

## 6  DISCUSSION

Based on previous context-adaptive methods (Minnen et al., 2018a; Lee et al., 2019), we have introduced a new entropy model for learned image compression. By combining global reference, we have developed a more accurate distribution estimating for the latent representation. Ideally, our combined entropy model effectively leverages both the local and global context information, which yields an enhanced compression performance. The positive results from global reference are somewhat surprising. We showed in Figure 7 that the combined entropy model provides a progressive improvement in terms of rate-distortion performance without increasing the complexity of the model.

Global reference model scans decoded latents and then finds the most relevant latent to assist the distribution estimating of target latent. Our reference model is inspired by recent works of reference-based super-resolution (Zhang et al., 2019; Yang et al., 2020). We extend this reference-based module in three ways to adapt it to image compression. First, we extend it to a single-image reference module. The second extension is that we incorporate the reference module into entropy model. The idea is that avoid influencing the highly compacted latent representation. Moreover, a confidence variable is introduced to enable adaptive reference. Intuitively, we can see how the local context and global reference are complementary as shown in Figure 9.

The improvement by the reference model also implies that current learned image compression is not ideal to model the spatial redundancy. The proposed reference model develops relevance across the latents of a single image. Reference within a single image limits its benefit. The upper part of latents has fewer decoded latents to reference. An alternative direction for future research may be to extend the reference model to multi-image compression. We also plan to investigate combining video compression with reference model to see if the two approaches are complementary.

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

# A  APPENDIX

## A.1  RESULTS ON THE CLIC VALIDATION DATASET

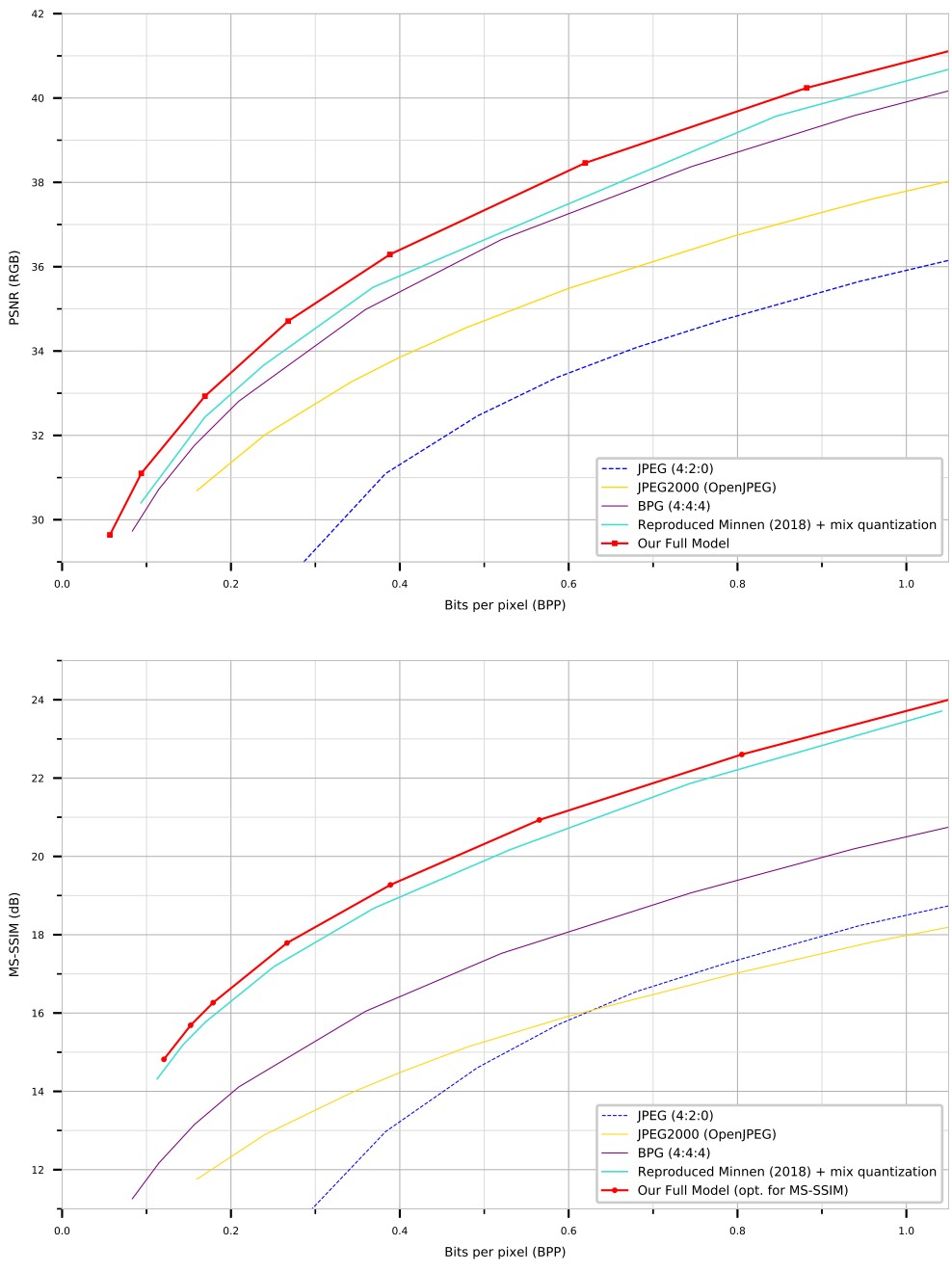

Figure 12:  Performance Evaluation on CLIC Validation dataset. Our method performs very well when optimized for MSE or MS-SSIM. Each point on the RD curves is calculated by averaging over the PSNR (or MS-SSIM) and bit rate for the 102 images from CLIC Validation dataset (`http://challenge.compression.cc/`).

| Encoder | Decoder | Hyper Encoder | Hyper Decoder | Context Model | Reference Model | Parameter Network |
|---|---|---|---|---|---|---|
| Conv: k5c192s2 | Deconv: k5c192s2 | Conv: k3c192s1 | Deconv: k5c192s2 | Masked: k5c384s1 | Masked: k3c384s1 | Conv: k1c1152s1 |
| GSDN | IGSDN | Leaky ReLU | Leaky ReLU | | | Leaky ReLU |
| Conv: k5c192s2 | Deconv: k5c192s2 | Conv: k5c192s2 | Deconv: k5c192s2 | | | Conv: k1c1152s1 |
| GSDN | IGSDN | Leaky ReLU | Leaky ReLU | | | Leaky ReLU |
| Conv: k5c192s2 | Deconv: k5c192s2 | Conv: k5c192s2 | Deconv: k3c768s1 | | | Conv: k1c768s1 |
| GSDN | IGSDN | | | | | |
| Conv: k5c384s2 | Deconv: k5c3s2 | | | | | |

Table 1: Each row corresponds to a layer of our generalized model. Convolutional layers are specified with the "Conv" prefix followed by the kernel size, number of output channels and downsampling stride (*e.g.* the first layer of the encoder uses 5×5 kernels with 192 output channels and a stride of 2). The "Deconv" prefix corresponds to upsampled convolutions, while "Masked" corresponds to masked convolution as in Van den Oord et al. (2016). GSDN stands for generalized subtractive and divisive normalization, and IGSDN is inverse GSDN. The three parameter networks share a similar architecture.

| Method | Encoder | Decoder | Hyper Encoder | Hyper Decoder | Context Model | Reference Model | Parameter Network |
|---|---|---|---|---|---|---|---|
| Minnen et al. (2018a) | 74.63 | 269.98 | 2.92 | 7.62 | 11.32 | NA | 8.15 |
| Ours | 84.12 | 279.47 | 2.92 | 7.62 | 11.32 | 7.70 | 24.46 |

Table 2: GFLOPs of each module for the proposed method and reproducing method (Minnen et al., 2018a). The size of the test image is 512×768.

| | 240p | 360p | 480p | 512×768 | 720p | 1080p | 4k |
|---|---|---|---|---|---|---|---|
| GFLOPs | 9.98 | 22.85 | 41.59 | 54.03 | 138.04 | 366.57 | 3296.99 |
| Proportion of Reference | 9.36% | 10.90% | 12.98% | 14.25% | 21.34% | 33.36% | 66.28% |

Table 3: GFLOPs of entropy model and proportion of reference model with various image size.

## A.2 ARCHITECTURE DETAILS

Details about the individual network layers in each component of our models are outlined in Table 1. The output of the last layer in encoder corresponds to the latents, and the output of the last layer in hyper-encoder corresponds to the hyper-latents. The output of the last layer in decoder corresponds to the generated RGB image. The three parameter networks share a similar architecture. The only difference is the number of the first layer's input channels in parameter network. The output of the parameter network must have exactly twice as many channels as the latents. This constraint arises because the entropy model predicts two values, the mean and deviation of a Gaussian distribution, for each latent.

## A.3 COMPUTATIONAL COMPLEXITY

Our main goal was to optimize for compression performance. To make a fair comparison with previous methods, we have taken care to match model capacities between Context+Hyperprior method (Minnen et al., 2018b) and Context-Adaptive method (Lee et al., 2019). We did not to choose a number of filters that would limit the capacities of encoder, decoder and entropy model.

Compared to Context+Hyperprior method (Minnen et al., 2018b), the increased computation by GSDN and global reference is about 10% when processing the 512×768 Kodak images. The complexity of search module is $O(N^2)$ about the size of latents, while the complexities of other modules are $O(N)$. Although not the main goal of our paper, the computational complexity is crucial for the application of deep learning method in industry field. We add two tables in appendix to describe computational complexity and FLOPs of our method. As you concerned, the complexity of reference model increases with image size more fast than other modules. The reason is that the search module computes similarity matrix throughout the latents. And we just use a naive way for the search module. So, there is some room to improve the global reference model and we are interested in researching this further.

We have not yet optimized our compression method for computational complexity. Rather, we chose the number of filters high enough to rule out the Reference model. We just set the model larger than necessary, and let the model determine the number of channels that yield the best performance.

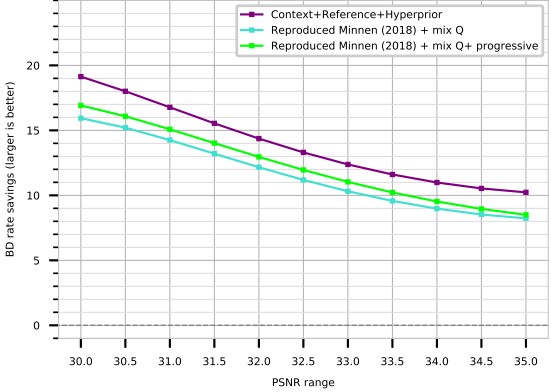

Figure 13: Each curve shows the rate savings at different PSNR quality levels relative to BPG. We reproduced similar models of (Minnen et al., 2018a) Context+ Hyperprior with two variants of mixed quantization and progressive entropy model. The training procedures (*e.g.*, training data, environment, hyper parameters, etc) are same with the other experiments.

Similar to reported in previous work, we found that small channels is enough and does not harm the compression performance when training model to low bit rates.

## A.4  VISUAL COMPARISON ON THE KODAK DATASET

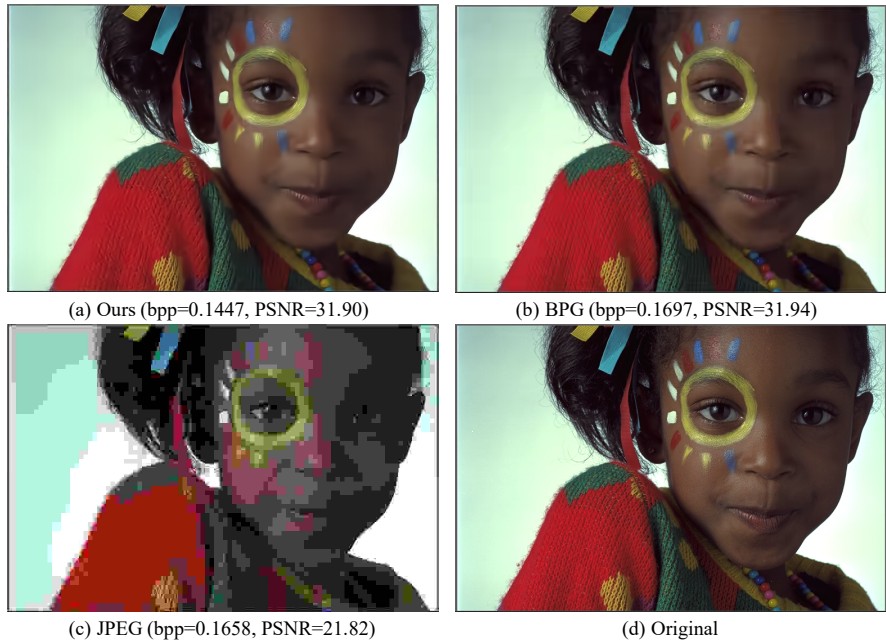

(a) Ours (bpp=0.1447, PSNR=31.90)   (b) BPG (bpp=0.1697, PSNR=31.94)

(c) JPEG (bpp=0.1658, PSNR=21.82)   (d) Original

Figure 14:  At similar bit rates, our combined method provides the highest visual quality on the Kodak 15 image. BPG introduces a few geometric artifacts around the girl's mouth and nose. JPEG shows severe blocking artifacts at this bit rate.

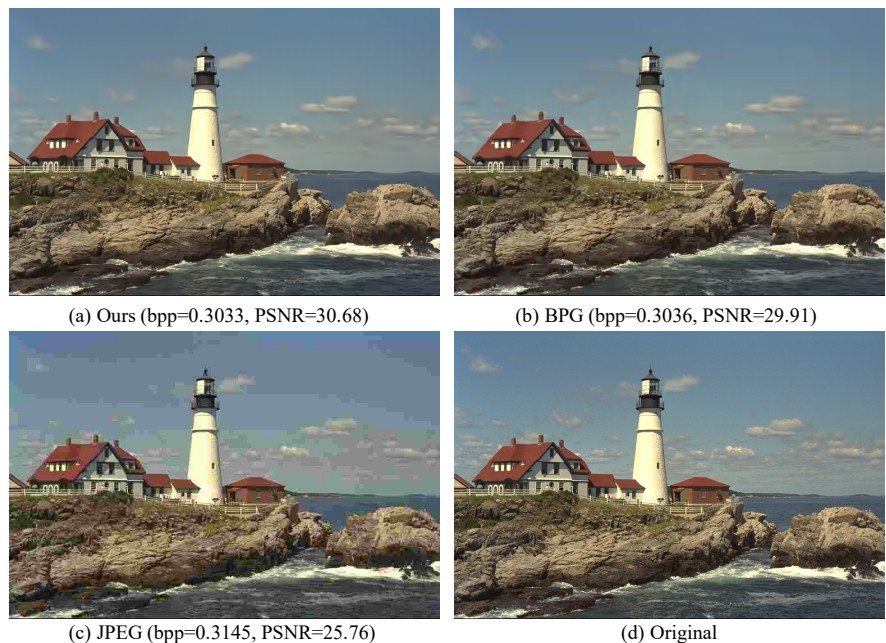

(a) Ours (bpp=0.3033, PSNR=30.68)  (b) BPG (bpp=0.3036, PSNR=29.91)

(c) JPEG (bpp=0.3145, PSNR=25.76)  (d) Original

Figure 15: At similar bit rates, our combined method provides the highest visual quality on the Kodak 21 image. BPG shows more "classical" compression artifacts, *e.g.*, ringing around the edge of the lighthouse.

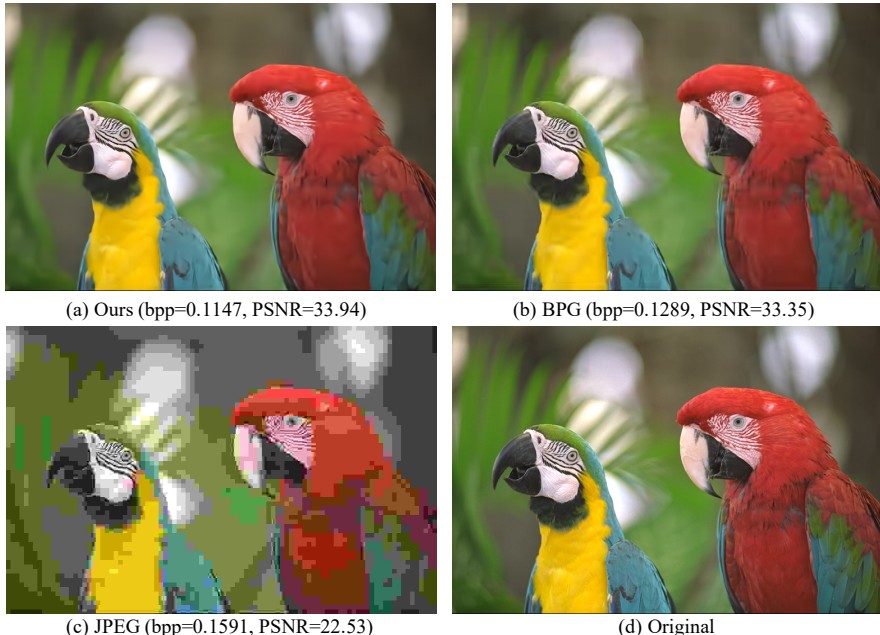

(a) Ours (bpp=0.1147, PSNR=33.94)  (b) BPG (bpp=0.1289, PSNR=33.35)

(c) JPEG (bpp=0.1591, PSNR=22.53)  (d) Original

Figure 16: At similar bit rates, our combined method provides the highest visual quality on the Kodak 23 image. Note that the BPG reconstruction has some ringing and geometric artifacts (*e.g.*, at the top of the red parrot's head).

## A.5 VISUAL COMPARISON ON THE CLIC VALID DATASET

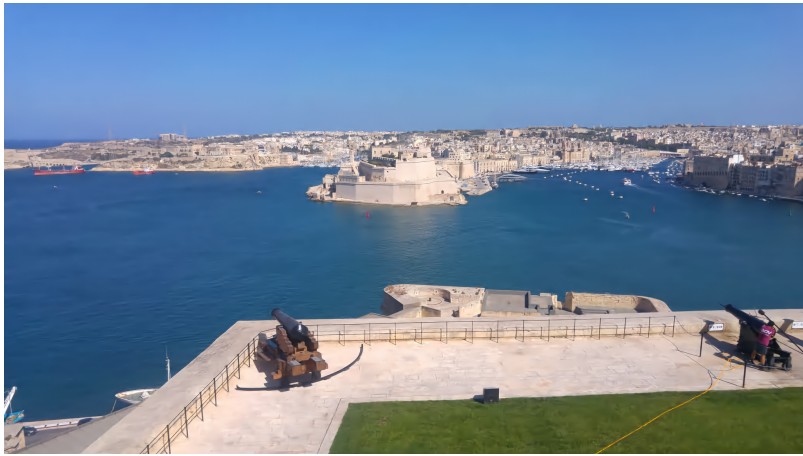

Ours (bpp=0.1512, PSNR=31.83)

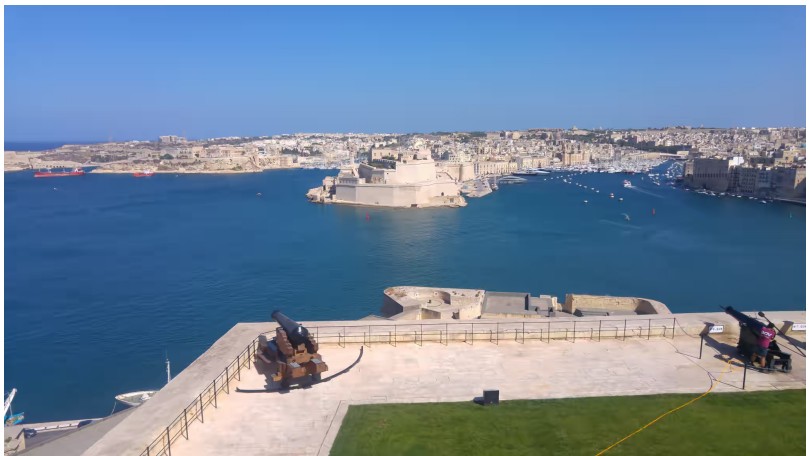

BPG (bpp=0.1678, PSNR=31.62)

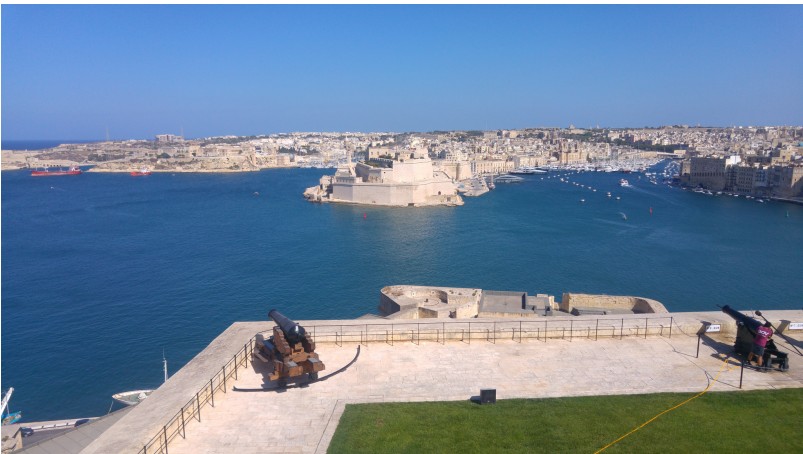

Original

Figure 17: At similar bit rates, our method provides the better visual quality on the high resolution image on CLIC valid dataset.

## A.6 ARCHITECTURE COMPARISON

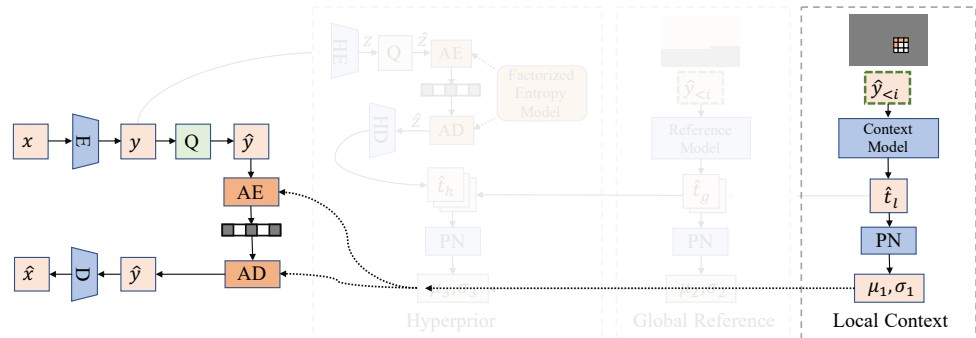

Figure 18: **Context-only Entropy Model.** This model relies only on an autoregressive process with a local context to predict the Gaussian parameters. The benefit of this approach is that no additional bits are added to the bitstream. The downside of this model is that it conditions predictions only on neighboring latents.

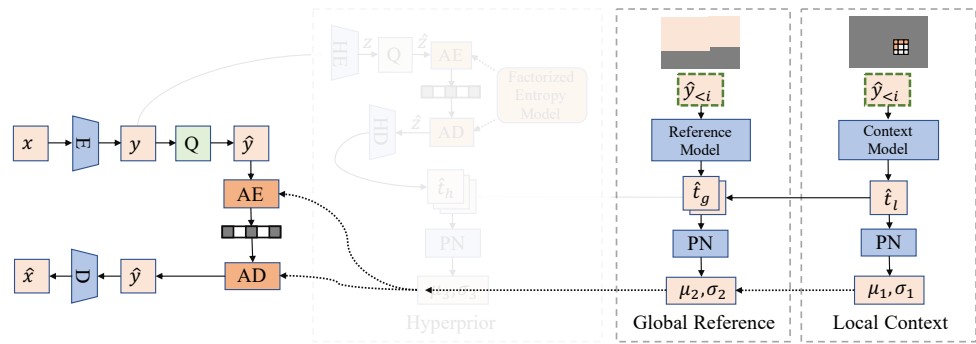

Figure 19: **Context + Reference Entropy Model.** This model combines global reference with local context. The benefit of this model is that it can access all previous latents.

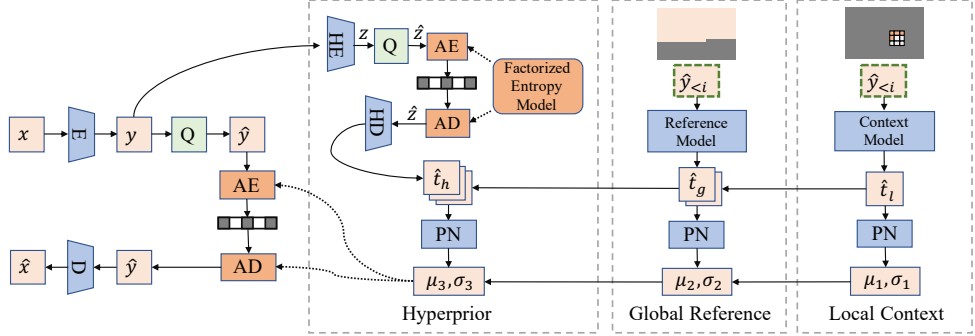

Figure 20: **Context + Reference + Hyperprior Entropy Model.** This model uses a hyper-network to learn a (hyper-)latent representation to transmit side information. The benefit of hyperprior in this model is that it learns to represent information useful for correcting the context-based predictions.

