# OpenReview forum: "Learning Accurate Entropy Model with Global Reference for Image Compression"
_ICLR.cc/2021/Conference — ICLR 2021 Poster_

### Official Review · AnonReviewer1 · 2020-10-26
**This is a good work, but I have concerns on the diversity of test examples, and computational complexity.**

**Rating:** 6
**Confidence:** 4

**Review:**

The authors introduce global reference into the entropy model for deep image compression.
They also develop a reference algorithm to ensemble local context, global reference and hyperprior.
This causes the algorithm to be robust to background noise.
Also, the authors develop GSDN module to handle mean-shifting issue.
The proposed method demonstrates good quality and memory usage gain.

This paper propose to take into account the global information as well as the local information to perform better image compression.
The authors also demonstrate comparison to popular image compression standards and recent deep learning approaches.
I think this work is a nice work, however I have two main concerns.
The dataset used for evaluation is rather outdated. Have the authors tried evaluating on recent image compression datasets, or custom data and compare with the state of the art?
Have the authors compared computational complexity? The main reasons why industry standards are not enthusiastic about deep learning approaches to compression is due to the computational complexity, not so much memory. Have the authors compared FLOPS? Moreover, since this work is dealing with global image information, it seems the complexity would increase rapidly with image size, while standard jpeg will relatively be not as severe. Have the authors experimented computational time with UHD, QHD, or 4k?

I am leaning towards accept but not by a lot.
I would like the authors to discuss upon
- Empirical results on more recent datasets
- Computational complexity and in terms of image size
- FLOPS
- Computational complexity and time with high resolution like UHD to 4k

After these comments, I would like to adjust the rating

---

> ### Author Response · Authors · 2020-11-20
> **Response to AnonReviewer1**
>
> ● Empirical results on more recent datasets.
>
> The results on the CLIC 2019 dataset is provided in the appendix.
>
> ● Computational complexity & FLOPs.
>
> Our main goal was to optimize for compression performance. To make a fair comparison with previous methods, we have taken care to match model capacities between Context+Hyperprior method (Minnen et al. 2018) and Context-Adaptive method (Lee et al. 2019). We did not to choose a number of filters that would limit the capacities of encoder, decoder and entropy model.
>
> Compared to Context+Hyperprior method (Minnen et al. 2018), the increased computation by GSDN and global reference is about 10% when processing the 512x768 Kodak images. The complexity of search module is O(N^2) about the size of latents, while the complexities of other modules are O(N). We add two tables in appendix to describe computational complexity and FLOPs of our method.
>
> We have not yet optimized our compression method for computational complexity. Rather, we chose the number of filters high enough to rule out the Reference model. We just set the model  larger than necessary, and let the model determine the number of channels that yield the best performance. Similar to reported in previous work, we found that small channels is enough and does not harm the compression performance when training model to low bit rates. The work (https://arxiv.org/abs/1912.08771) has analyzed for computationally efficient deep leanrning based image compression.

---

### Official Review · AnonReviewer2 · 2020-10-28

**Rating:** 6
**Confidence:** 4

**Review:**

This paper propose two methods for improve deep image compression performance: (i) Global Reference Module and (ii) Mean-shifting GDN Module (GSDN). (i) Global Reference Module searches over the decoded latents to find the relevant latents to the target latent for improve accuracy of entropy estimate. Authors extended Yang et al. 2020 method to using masked patch. (ii) GSDN extends GDN to use subtractive operation.

Pros:
- Proposal seems better Rate-Distortion results than Lee 2019 and Minnen 2018 (Figure 6 and 7).

Cons:
- The method for generating U (2D feature maps) is not clearly described; it is unclear how the output channel of the parameter network (768) is calculated in the form of 2D feature maps.
- In Table 1, the output channel of the Encoder is 384, while the corresponding input channel of the decoder is 192. I couldn't understand why the number of channels are not the same.
- In Figure 5, the meaning of σ's log is unclear and seems not appear to have been mentioned in the text.
- The proposed method uses the mix quantization approach (Minnen & Singh 2020), but the evaluation of figure 7 is compared to Minnen 2018 as Context + Hyperprior so it is not fair comparison. For example, to put the results of the Minnen 2018 approach + mix quantization on it and compare it will make the claim of the proposal effect credible.
- According to Figure 7, GSDN appears to be effective. It is an interesting, but I thought the effects were less explained. It would be more convincing if direct data on the mean-shifting problem were presented.

---

> ### Author Response · Authors · 2020-11-20
> **Response to AnonReviewer2**
>
> ● The method for generating U.
>
> We use the context model to predict the Gaussian parameters (i.e. \mu_1 and \sigma_1) of latents solely. The latents y_hat are now modeled as Gaussian with mean \mu_1 and standard deviation \sigma_1. The probabilities of the latents are then calculated according to \mu_1 and \sigma_1 as in eq.(2). As reference model is designed in spatial dimension, confidence U is obtained by averaging the probabilities across channel.
>
> We hope that the current revision is much clearer.
>
> ● The channel setting of encoder and decoder.
>
> The channel number in Table 1 denotes the layer's output channel for short description. That means the input channel of the first deconv in decoder is 384, while the output channel of the first deconv in decoder is 192. We have fixed this unclear presentation.
>
> ● The meaning of \sigma's log.
>
> It is the Log-Sum-Exp trick for resolving the under- or overflow issue.
>
> ● The results of the Minnen 2018 + mix quantization.
>
> We have reproduced the Minnen 2018 + mix quantization method and showed the results in appendix. The training procedures (e.g. training data, environment, hyper parameters, etc) are same with the other experiments. Although benifiting from the mix quantization, the results of the reproduced model (the Minnen 2018 + mix quantization) is comparable to the results report in their work. Note that we trained all the models on CLIC training dataset which has 1631 images. Context+Hyperprior method (Minnen et al. 2018) has no claim about the training set, while their previous work used 1 million images. Context-Adaptive method (Lee et al. 2019) used 32420 images. The gap may lie in the difference of training data.
>
> ● GSDN.
>
> We have added some supporting visualizations of GSDN. Compared to GDN, GSDN relieves the mean-shifting problem of the latents. Figure.10 shows the histograms of the latent representation by GDN-based model and GSDN-based model. Each plot corresponds to one channel of the latents over 24 Kodak images. Four channels with highest entropy are visualized.
>
> The ides of GDSN is inspired by the zero-mean definition of Gaussian density. And in the orignal work GDN, mean
>  removal is necessary for preprocessing. On the other hand, we think that GDN has the potential to subsume the effects of batch normalization. It is intuitive to apply a subtractive operation in GDN.

---

> > ### Comment · AnonReviewer2 · 2020-11-21
> > **Thank you for updating**
> >
> > I think the update makes the author's point clearer. I have updated the rates.

---

### Official Review · AnonReviewer3 · 2020-10-28
**Interesting paper taking into account the global context to efficiently decrease the compression rate for image compression**

**Rating:** 7
**Confidence:** 3

**Review:**

**Summary of paper**
The paper presents a learning-based approach for image compression. To reduce the compression rate, it describes two novel extensions, one to take the global context into account and an improved version of the commonly used GDN layer. Their advantage has been shown in a thorough ablation study. Overall, the method achieves superior performance compared to standard codecs as well as other state-of-the art learning-based method on the evaluated dataset (from Kodak).

**Strengths**

(S1) The approach is clearly described, and the figures help to follow the paper.

(S2) The proposed approach of using a reference model to consider the global context is novel for the application of image
compression.

(S3) Suggestion to improve GDN layer to address the mean shift problem of the existing formulation.

(S4) The ablation study shows the effect of the different modules.

(S5) The method achieves superior results on the tested dataset (Kodak), standard in image compression, and both PSNR/SSIM have been reported.

**Weaknesses**

(W1) Runtime for encoding and decoding not listed. In case, that the global reference model leads to some overhead in runtime, especially for decoding time, that would be worth mentioning. Also are there any limits in terms of image size the method can handle?

**Justification**
Interesting approach with superior results.

---

> ### Author Response · Authors · 2020-11-20
> **Response to AnonReviewer3**
>
> ● Computational complexity.
>
> Our main goal was to optimize for compression performance. To make a fair comparison with previous methods, we have taken care to match model capacities between Context+Hyperprior method (Minnen et al. 2018) and Context-Adaptive method (Lee et al. 2019). We did not to choose a number of filters that would limit the capacities of encoder, decoder and entropy model.
>
> Compared to Context+Hyperprior method (Minnen et al. 2018), the increased computation by GSDN and global reference is about 10% when processing the 512x768 Kodak images. The complexity of search module is O(N^2) about the size of latents, while the complexities of other modules are O(N). We add two tables in appendix to describe computational complexity and FLOPs of our method.
>
> We have not yet optimized our compression method for computational complexity. Rather, we chose the number of filters high enough to rule out the Reference model. We just set the model  larger than necessary, and let the model determine the number of channels that yield the best performance. Similar to reported in previous work, we found that small channels is enough and does not harm the compression performance when training model to low bit rates. The work (https://arxiv.org/abs/1912.08771) has analyzed for computationally efficient deep leanrning based image compression.
>
> ●  The limits of image size.
>
> As we have reported computational complexity in terms of image size (Table 3 in Appendix), the most computation lies in encoder and decoder. There is GPU memory limit when processing high-resolution image. The limit mainly depends on the large feature map in the first layer of encoder or the last layer of the decoder. Similar to reported in previous work, we found that small channels is enough and does not harm the compression performance when trainning model to low bit rates. In industry application, efficient network would ease the limits of image size.

---

### Official Review · AnonReviewer4 · 2020-10-29
**This paper transferred the reference-based method in Super-Resolution to help remove the global spatial redundancy in the entropy model for image compression and achieved performance improvement. Generally, the motivation is clear and reasonable. However, some details about the model, experiments, and statements seem to be missing, which makes the whole paper somewhat unconvincing.**

**Rating:** 5
**Confidence:** 5

**Review:**

1	The explanation of the proposed “confidence U” is significantly deficient. Further statements to the calculation process, theory and function are expected. Please provide more details.
2	What the point of the operation that the output of global reference is multiplied by similarity S and confidence U directly as shown in Figure 5?  Will the direct multiplication by the confidence U impair the function of the attention S? A more reasonable and efficient method is expected for applying the confidence map.
3	Different from conventional entropy models, this paper proposed to progressively use the prior from context model, reference model and hyperprior model instead of obtaining priors respectively and then combining them to obtain estimation results. But the paper lacks the explanation and experiments to the rationality of the design of the progressive process. The comparison results of whether progressive or not are missing.
4	Generally, the performance gain is very limited as shown in the RD curves in Figure 6. In particular, why the performance gain appears to be so minor in the low bitrate range?
5	In Figure 10, why the comparison results with any learned compression methods are missing? The comparisons with various methods only on Kodak dataset in Fig. 6 are not convincing.
6	The visual comparison results on the CLIC dataset of high resolution are missing. And some key references for deep-network based image compression are also missing.

---

> ### Author Response · Authors · 2020-11-20
> **Response to AnonReviewer4**
>
> ● The statements of "confidence U".
>
> We have rewritten the part about "confidence U" sections in order to improve clarity. We hope that the current revision is much clearer. Please let us know if there are any other (or new) parts which you find hard to read, we are happy to make further improvements.
>
> ● The point of the operation for confidence U and similarity S.
>
> Our entropy model provides coarse-to-fine estimations for Gaussian parameters in a progressive way. The first stage (i.e. context model) generates a coarse estimation, which is refined by offset and scaling generated by the next two stages. It is not appropriate to apply all the refinements equally into the estimations. The confidence U and similarity S are used to softly apply the refinements from the reference model. It is intuitive to multiply the refinements by U and S, which has proven efficient in reference-based SR work.
>
> The multiplication by the confidence U would influence the similarity S. The U map represents the texture score of the reference latent, which measures the complexity of the reference feature. It agrees with what we have assumed that U would compensate for the weakness of S. We have visualized a example of S and U in revision. As shown in Figure.11, the S map is tends to represent shape while the U map is tends to represent texture, e.g., the wood pile on the right of the boat.
>
> However, we agree that the confidence map could be applied more efficiently. We will likely do more research to explore a more efficient way to apply the confidence map and the similarity map in the future.
>
> ● The point of progressive process.
>
> Thank you for pointing this out. We have constructed a Context+Hyperprior+Preogressive model and provide the results in the appendix. There is minor benefit coming at the progressive design.
>
> As for the point of "progressively use the prior from context model, reference model and hyperprior model instead of obtaining priors respectively and then combining them to obtain estimation results", we tried both ways when reference model is introduced without confidence U. As we mentioned in the above problem, we then introduced confidence U to compensate for the weakness of similarity S in reference-based entropy  model. Since U is generated by the context model, the progressive way is intuitive and necessary for our entropy model.
>
> ● The explanation for the Figure 6.
>
> The RD-curve visualization in Figure.6 provides a numerical perspective. The rate savings (Figure.7) visualization highlights how the rate savings varies with quality (and thus bit rate) in a more readable way than a standard RD graph. The performance gain is greater in the low bitrate than that in the high bitrate.
>
> ● The comparison results with other learned methods for CLIC valid.
>
> We agree this would be desirable, but this is limited in practice as other learned methods have no experiments for CLIC valid. As the results for other learned methods on CLIC valid is missing, we have reproduced Minnen (2018) with mix quantization for a comparison. We have added the results in appendix. Due to the short of time, only the PSNR results are provided. The MS-SSIM results would be provided in further revision.
>
> ● The visual comparison results on the CLIC dataset of high resolution.
>
> It is added in the appendix.
>
> ● References for deep-network based image compression.
>
> Thanks, we have added some references of learned image compression. Please let us know if there are any other works missing.

---

### Author Response · Authors · 2021-01-25
**Updated revision**

Hello everyone,

Thank you for your detailed critique of our paper. We have worked hard to revise our paper and address all of the points you have raised.

Thank you - the authors.

---

### Decision · Program_Chairs · 2021-01-07
**Final Decision**

**Decision:**

Accept (Poster)

**Comment:**

This paper received moderately good reviews, 3 positives (6, 6, 7) and 1 negative (5). The reviewers are generally positive about the main idea but identified several limitations; performance improvement is marginal compared to existing approaches, the proposed method incurs higher computational complexity, and the presentation is not clear enough. Some of these issues are addressed in the rebuttal, though. Overall, the merits of this work outweigh the drawbacks and I recommend accepting this paper.